# Trade Flow Optimization Model for Plastic Pollution Reduction

**DOI:** 10.3390/ijerph192315963

**Published:** 2022-11-30

**Authors:** Daming Li, Canyao Liu, Yu Shi, Jiaming Song, Yiliang Zhang

**Affiliations:** 1Department of Physics, Yale University, New Haven, CT 06511, USA; 2Yale School of Management, Yale University, New Haven, CT 06511, USA; 3Department of Computer Science, Stanford University, Stanford, CA 94305, USA; 4Department of Biostatistics, Yale University, New Haven, CT 06511, USA

**Keywords:** network flow model, plastic pollution, optimization

## Abstract

Managing plastic waste from an international perspective is complex, with many countries in the trade network playing distinct roles at different stages of the life-cycle of plastics. Trade flows are therefore the key to understanding global plastic market and its supply chains. In this paper, we formulate an optimization problem from the perspective of reducing global ocean plastic pollution, and create a novel framework based on a network flow model to identify the optimal international trade flows over the life-cycle of plastics. Our model quantifies global flows of production, consumption, and trade across the life-cycle of plastics from raw inputs and subsequent plastic products to its final stage as waste. Using panel data on plastic consumption, waste, and production, we compare the trade flows in reality and the optimal trade flows determined by our model and find that the two are highly correlated. We highlight the policy implications based on our model: increasing trade capacity and improving recycle rates in developing countries.

## 1. Introduction

Plastics are one of the defining materials in modern human society, with their presence in almost all sorts of products in everyday life. However, mismanagement of plastic waste can cause serious environmental pollution, notably in the form of debris that is dragged by rivers into the ocean [1,2,3,4]. Dealing with plastic waste mismanagement turns out to be challenging. On the one hand, the entire life-cycle of conventional fossil-based plastics has multiple phases: from feedstocks, to primary plastics in resin pellet and fiber forms, through to intermediate plastic goods, final manufactured plastic goods and plastic waste. On the other hand, the value chain of the plastics industry is very complex, with many countries in the high-volume trading network playing distinct roles at each one of the stages of the life-cycle of plastics [5,6,7,8,9]. International plastics trade flows are therefore the key to understanding global plastic market and its impact on oceanic pollution. Given the complexity and high-volume of data, there is need for a quantitative model from an operational perspective, to help understand how different policy levers impact global plastic pollution, and to assess and improve existing international trade flows and policies. To our knowledge, such a quantitative modeling framework is still missing in the field. In this paper, we formulate an optimization problem from the perspective of reducing global ocean plastic pollution, and create a novel framework based on a network flow model to identify the optimal international trade flow patterns over the life-cycle of plastics.

To that end, we first study the distributional patterns of raw inputs for plastic production, plastic products themselves, and plastic waste, as well as their relationships with a country’s other characteristics such as GDP or income level (Section 3.1). We establish four facts at the country level: (1) net exports of plastic products are positively correlated with oil production (key input); (2) high-income countries tend to be net importers of plastic products whereas low-income countries tend to be net exporters; (3) high-income countries tend to be net exporters of plastic waste whereas low-income countries tend to be net importers; (4) plastic recycle rate is positively correlated with a country’s GDP per capita and mismanaged plastic waste rate is negatively correlated with GDP per capita. Those empirical findings motivate our assumptions over the model.

Next, we construct a network flow model featuring the life-cycle of plastic in both domestic and international economic activities (Section 2.2). Our model quantifies global flows of production, consumption, and trade across the life-cycle of plastics—from raw inputs and subsequent plastic products to its final stage as waste. Crucially, we include international trade of plastic waste for recycling, where countries can import plastic waste for the sole purpose of it being used in production and consumption. Our model describes five stages of plastic flow: (I) from fossil fuel to raw plastic materials; (II) from materials to products; (III) from products to plastic waste via consumption; (IV) from plastic waste to mismanaged plastic waste, and (V) from plastic waste to recycled raw plastic materials. We include the effects of international trade at stages I, II, and IV, and quantify pollution to the environment using cost functions at stage IV.

Given the natural and imposed constraints over the model, we wish to search among the plastic network flows that encompass assignments of plastic production, consumption, trade, and recycling for countries across the globe. Within polynomial time, we can find an optimal plastic flow with minimum cost, i.e., a global assignment that minimizes the global (or ocean) environmental impact while ensuring the needs of consumption in every country. Our assignment is guaranteed to reduce global (or ocean) pollution over cases where trading recyclable plastic waste does not occur.

We compare the trade flows in reality and the optimal trade flows determined by our model and find that the two are highly correlated (Section 3.2), indicating that our model is fairly close to reality. Next, we further compare our model solutions under different parameter settings and offer two policy insights.

First, increasing trade capacity of plastic waste reduces pollution (Section 4.1.1). Specifically, if we increase the current trade capacity between countries with plastic waste trade by 10×, we can reduce the global mismanaged plastic waste by 1.4% and the ocean plastic pollution by 13.8%. There are also differential effects of lifting plastic waste trade capacity for different countries, with higher improvements for countries with lower GDP per capita. For Philippines, we find that the above policy reduces its ocean plastic pollution by 90.8%, and reduced its share of ocean plastic waste from 14.4% to 1.5%.

Second, increasing the plastic waste recycle rate reduces pollution (Section 4.1.2). This is not a surprising finding per se, but we also observe a clear heterogeneity in its effects across countries and regions. Improvement in plastic recycling capabilities in developing countries tend to help more with this global issue, while that in developed countries is relatively marginal. In addition, larger developing countries such as China have larger policy lever and its improvement in recycling can significantly alleviate the global pollution. Therefore, it is important that developed countries provide more technology assistance to developing countries in improving their recycling system.

In summary, we develop a novel model for optimizing and evaluating international trade flows over the life-cycle of plastics. Our flexible framework, combined with real-world data, can provide lots of actionable insights for governments’ trade policies across the globe.

## 2. Materials and Methods

### 2.1. Data

We utilize online data sources to support our analysis. They are:Plastic trade flow: We download the country-to-country trade flow of different plastic products from the World Bank database.Country code: We download the alpha-3 code and numeric code for each country (Link for data download: https://www.iban.com/country-codes (accessed on 12 November 2021), [IBAN:country codes alpha-2 and alpha-3]) in order to aggregate the information from different data sources.Oil production by country: We download the oil production data for different countries in 2010 from the “Our World in Data” website (Link for data download: https://ourworldindata.org/fossil-fuels (accessed on 12 November 2021), [Our World in Data: Oil production]). We use this to reflect each country’s capacity for producing raw materials for plastic production. This is used to estimate capacities in Stage I of our model.Industrial GDP by country: We download the GDP of the industrial sector by country from Statistics Times website (Link for data download: https://statisticstimes.com/economy/countries-by-gdp-sector-composition.php (accessed on 13 November 2021), [Statistics Times: Industrial GDP]). We use this to reflect the production of plastic products in each country. This is used to estimate capacities in Stage II of our model.Aggregate consumption by country: We download the aggregate consumption data from the World Bank database (Link for data download: https://data.worldbank.org/indicator/NE.CON.TOTL.ZS (accessed on 13 November 2021), [World Bank: Aggregate consumption]). We use this to reflect the demands of plastic products in each country. This is used to estimate capacities in Stage III of our model.Municipal solid waste recycle rate by country: We download data for a selected list of countries from Statista website (Link for data download: https://www.statista.com/statistics/1052439/rate-of-msw-recycling-worldwide-by-key-country/ (accessed on 13 November 2021), [Statista: Municipal solid waste recycle rates]). We use this as a proxy for plastic waste recycle rate. This is used to estimate capacities in Stage V of our model.

After gathering and downloading the data, we carry out preprocessing for the following data:Plastic trade data: We use the same year of the world trade data and the data to aggregate. For example, in our trade flows optimization model, we use the 2010 world trade data because the mismanaged rate data are measured in 2010 [2]. We use the average trade volume of the import of country A from country B and the export of country B to country A as the final trade volume of the specific form of the plastic from country B to country A. The final trade volume between country A and country B is further used as the parameter for trade capacity in the trade flows optimization model.Mismanaged plastic waste data: We use the average mismanaged plastic waste rate of the countries with available data to impute the mismanaged plastic waste rate of the countries without the information. Mismanaged plastic waste rate is defined as the proportion of total plastic waste generation.Plastic waste recycle rate: Because we only have recycle rates for key countries in the data downloaded (32 countries in total), we impute the recycle rates by first fitting a regression of recycle rate on GDP per capita for the countries we have data for and then getting the predicted values for all countries. A practical justification is provided in Section 3.1.

The above data can evolve over years, with the global plastics production steadily increasing (source: https://ourworldindata.org/grapher/global-plastics-production (accessed on 13 November 2021), [Our World in Data: Global plastics production]) and similarly for the projected mismanaged plastic waste [2]. Still, the relative shares in plastics production, consumption, and recycling capabilities across countries are relatively stable (source: https://stats.oecd.org/index.aspx (accessed on 13 November 2021), [OECD.Stat database (environment chapter)]). Although, here, we relied on 2010 data to constrain the model, the framework is generally applicable to any arbitrary year.

### 2.2. Model

In this and the next subsections, we provide the technical details of our modeling methods. We describe a method used to find optimal production, trade, and recycling allocations that minimizes certain metrics of plastic pollution. Our method is related to the concept of network flow problems in optimization theory [10,11,12]; we optimize for the flow that describes the allocations of production, trade, and recycling, while satisfying constraints on production, trade, recycling, and consumption based on real-world data (see Figure 1).

#### 2.2.1. Stages of Plastic Flow and Problem Statement

We consider the life-cycle of a unit of plastic particle that enters and leaves the world economy via the following stages:Stage IRaw forms of plastic are created from fossil fuel and related precursors; this stage mostly depends on the countries’ fossil fuel supply. International trade can happen at this stage to balance supply and demand.Stage IIManufactured goods that contain plastic as a component are created from raw forms of plastic, possibly via intermediate forms of plastics; this stage mostly depends on the countries’ industrial production capacity. International trade can happen at this stage to balance supply and demand. We note that this also considers products that have plastic as a component but are not primarily plastic.Stage IIIPlastic waste is produced as a result of consumption of manufactured goods; this stage mostly depends on the countries’ consumption capacity.Stage IVA part of the plastic waste is managed domestically, and mismanaged plastic waste (MMPW) is produced as a result; this stage depends on the countries’ ability to collect and handle plastic waste properly (such as by recycling).Stage VThe remaining plastic waste is managed through international trade, where the importer of plastic waste will either recycle the plastic for its own use or dispose of it (which may lead to increased pollution from the importer). This stage is subject to supply, demand, and certain domestic policies (such as tariffs and import bans). We assume that the recycled plastic particles will become raw forms of plastic (Stage I) that can be reused in production.

International trade happens at Stages I, II, and V. We are interested in designing trade policies happening at these stages that benefit the environment while keeping our standard of living. To this end, we make two assumptions to our model:Assumption IOur trade policies will not reduce consumption of manufactured goods at Stage III.Assumption IICountries will not import plastic waste at Stage V unless said waste will be used for recycling.

#### 2.2.2. Network Flow Model of Plastics

Now, we are ready to describe our method as solving a minimum-cost maximum-flow problem on a directed acyclic graph (DAG) G=(V,E). The nodes (vertices) V represent the countries at certain stages and edges E represent the transition to certain stages (production, consumption, trade, or waste disposal). Edges additionally have two properties: capacity, which describes an upper limit for the possible plastic flow through the edge (e.g., production, consumption, or trade), and cost, which describes the environmental cost efficiency (we also use the term “cost” for conciseness) of the edge (e.g., waste disposal). We note that the cost efficiency is over each unit of plastic; if the cost efficiency for one country’s waste disposal is 0.5, then we mean that one unit of plastic waste will generate 0.5 unit of mismanaged plastic waste.

Let *n* be the total number of countries in the world (indexed 1 to *n*). In our network, plastics are categorized by their life cycle. We use the notations $A_{i},B_{i},C_{i},D_{i},E_{i}$ to denote nodes of a country indexed *i* for each respective stage. The nodes and edges are defined as follows:Stage IFirst, we define *n* “production” edges from a source node *S* to every $A_{i}$, with cost being small and capacity depending on the countries’ capacity to produce raw plastic materials. Then, we define n(n−1) “trade” edges between $A_{i}$ and $A_{j}$, with cost being small and capacity depending on the export limit of raw plastic materials from country *i* to country *j*.Stage IIFirst, we define *n* “production” edges from $A_{i}$ to $B_{i}$, with cost being small and capacity being the industrial capacity to create products from the raw plastic materials. Then, we define n(n−1) “trade” edges between $B_{i}$ and $B_{j}$ (i≠j), with cost being small and capacity depending on the export limit of products from country *i* to country *j*.Stage IIIWe define *n* “consumption” edges from $B_{i}$ to $C_{i}$, with cost being zero and capacity being the consumption capacity of country *i*. There is no trade at this stage.Stage IVWe define *n* “waste” edges from $C_{i}$ to $D_{i}$, with cost depending on the rate of creating mismanaged plastic (MMPW) and capacity being infinite. This stage models the process of plastic waste being discarded, as well as plastic waste that is not recovered during the recycling process. We can modify the cost to optimize different metrics of plastic pollution, such as total MMPW on the planet, in a specific country, or the total MMPW that results in ocean pollution. Following common practices in network flow definitions, we also define *n* edges between each $D_{i}$ and a sink node *T* with zero cost and infinite capacity. There is no trade at this stage.Stage VWe define $n^{2}$ “recycle” edges from $C_{i}$ to $E_{j}$ (*i* can be equal to *j*) with cost being zero and capacity being infinity. We further define *n* edges between $E_{j}$ to $A_{j}$ for each country *j*, with cost being zero and capacity being the country’s maximum amount to recycle plastic. This naturally also sets an upper limit to the capacity over the “recycle” edges.

Given the definition of the DAG, we can see the life-cycle of a unit of plastic via a path from *S* to *T* (Figure 2)

Type I: Not recycled. The unit of plastic is produced, consumed, and discarded in country *i*.
$$S\overset{\mathrm{raw}}{\to }A_{i}\overset{\mathrm{product}}{\to }B_{i}\overset{\mathrm{consume}}{\to }C_{i}\overset{\mathrm{waste}}{\to }D_{i}\to T$$Type II: Recycled. The unit of plastic is produced and consumed in country *i*, but recycled by country *j* via international trade; then, it is produced, consumed, and discarded in country *j*.
$$S\to A_{i}\to B_{i}\to C_{i}\overset{\mathrm{traded}}{\to }E_{j}\overset{\mathrm{recycled}}{\to }A_{j}\to B_{j}\to C_{j}\to D_{j}\to T$$We note that this path can be extended as long as there are countries willing to recycle the unit of plastic from *j*.

Here, we see that in Type II, one unit of plastic satisfies one unit of consumption in both countries *i* and *j*, which is impossible in Type I. Our Assumption II also ensures that the traded plastic waste will be recycled and used again in consumption. For future reference, we define the term “Total Flow” as the total amount of plastic products that are not recycled.

**Figure 2 ijerph-19-15963-f002:**
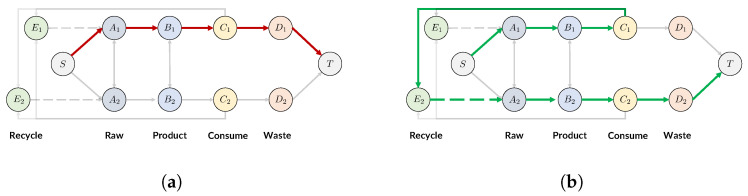
The flow of one unit of plastic; the plastic can be either discarded directly (**a**, red path) or recycled (**b**, green path). With recycling, one unit of plastic can satisfy two units of consumption. (**a**) Type I: Not recycled; (**b**) Type II: Recycled.

#### 2.2.3. Objective Function

We can define an objective over the trade assignments in Stages I, II, and V; as well as production, consumption, and discard stages. We describe this as a non-negative flow function $f:E\to R_{\geq 0}$ over edges. We can then optimize the flow function *f* to reduce plastic pollution. Our goal is to minimize the amount of pollution, defined via the cost in the edges:(1)$$\begin{array}{c} L\left(f\right):=\sum_{e\in E}^{n}f\left(e\right)\;\cdot \;\mathrm{cost}\left(e\right) \end{array}$$
where cost:E→R is a cost function that captures a measure of pollution on each unit of plastic through the edge (e.g., cost of discarding one ton of plastic), which can be the mismanaged plastic waste rate. The feasible solutions should satisfy the following constraints:Constraint IFlow does not exceed capacity, i.e., ∀e∈E,f(e)≤cap(e), where cap:E→R is the capacity function that we defined to model the constraints at each edge. The capacity is the trade volume of the corresponding form of plastic between the countries.Constraint IIExcept for *S* and *T*, every node has same amount of inflow and outflow. $\forall v\in V\backslash \left\{S,T\right\},\sum_{e\in E}f\left(e\right)I\left(v,e\right)=0$, where I:V×E→{−1,0,+1} is a function that takes value +1 if *v* is the receiving node of *e*, −1 if *v* is the sending node of *e*, and 0 if *v* is not a node on the edge *e*.Constraint IIIConsumption capacities are satisfied for every country, i.e., $\forall i\in \left[n\right],f\left(B_{i}\to C_{i}\right)=\mathrm{cap}\left(B_{i}\to C_{i}\right),$ where instead of the usual inequality constraints, here we use the equality constraint.

We note that this defines a linear program over *f* where methods such as interior point algorithms can be applied. To maximize computational efficiency of the optimization problem, however, we leverage algorithms on minimum-cost maximum flows that are guaranteed to give exact optimal solutions in polynomial time for integer cost and constraints (when costs and constraints are floats, multiplying them with a large number and then converting to integer often suffices), as well as a binary search method to find the optimal flow $f^{⋆}$ that minimizes impact on the environment while satisfying consumption needs (details in Section 2.4.1).

### 2.3. Parameters of the Plastic Flow

We can simulate different scenarios (e.g., trade limits) by setting different capacity functions cap and cost functions cost. As our interest is primarily on the effect from plastic waste trade, we set the following edge values for all our experiments (unless specified, costs are zero (as we mostly focus on plastic pollution) and capacities are infinity):The total consumption $F_{\mathrm{tot}}$ over the world is derived from yearly plastic production, measured in tons per year (source: https://ourworldindata.org/grapher/global-plastics-production (accessed on 13 November 2021), [Our World in Data: Global Plastics Production]).The capacities of countries’ supply of raw plastic materials (Stage I) are distributed according to oil supply in each country, i.e., $\mathrm{cap}\left(S\to A_{i}\right)=\left\{\mathrm{oil}\;\mathrm{supply}\;\mathrm{of}\;i\right\}\times F_{\mathrm{tot}}/\left\{\mathrm{global}\;\mathrm{oil}\;\mathrm{supply}\right\}$; the sum of all supplies is $1.2F_{\mathrm{tot}}$.The capacities of countries’ production of products that consist of plastics (Stage II) are distributed according to industrial GDP in each country, i.e., $\mathrm{cap}\left(A_{i}\to B_{i}\right)=\left\{\mathrm{industrial}\;\mathrm{GDP}\;\mathrm{of}\;i\right\}\times F_{\mathrm{tot}}/\left\{\mathrm{global}\;\mathrm{industrial}\;\mathrm{GDP}\right\}$; the sum of all production is $1.1F_{\mathrm{tot}}$.The capacities of countries’ consumption of products (Stage III) are distributed according to (nominal) GDP in each country, i.e., $\mathrm{cap}\left(B_{i}\to C_{i}\right)=\left\{\mathrm{GDP}\;\mathrm{of}\;i\right\}\times F_{\mathrm{tot}}/\left\{\mathrm{global}\;\mathrm{GDP}\right\}$; the sum of all consumption is $F_{\mathrm{tot}}$.The capacities of countries’ disposal of products (Stage IV) are infinite, but the cost is mismanaged plastic waste rate, i.e.,
$$\begin{array}{c} \mathrm{cost}(C_{i}\to D_{i})=\\ \{\mathrm{mismanaged}\;\mathrm{plastic}\;\mathrm{waste}\;\mathrm{generation}\;\mathrm{of}\;i\}/\{\mathrm{plastic}\;\mathrm{waste}\;\mathrm{generation}\;\mathrm{of}\;i\}. \end{array}$$We use the average mismanaged plastic waste rate to impute the countries without the data.

Our main focus is on international trade of plastic waste, which is reflected in the capacities of the following edges:Trade capacities. The capacities over countries’ trade of plastic waste $\mathrm{cap}(C_{i}\to E_{j})$ (Stage V) are proportional to the real-world trade data in metric tons. The details about how we obtain the trade volume between the countries are described in Section 2.1. These can be influenced more directly by a country’s trade policies.Recycling capacities. The capacities over countries’ ability to recycle plastic waste $\mathrm{cap}(E_{j}\to A_{i})$ (Stage V) are proportional to the countries’ GDP per capita. These depend more on the technology level of the country, and are less sensitive to trade policies.

To control the capacities, we often multiply a constant hyperparameter over all the base capacities derived from data. We may also increase or decrease these capacities for individual countries to simulate different trade scenarios.

### 2.4. Experimental Details of the Model

#### 2.4.1. Binary Search Method for Finding Optimal Flow Solution

First, we leverage the following observation:TotalConsumption=TotalFlow+TotalRecycling,
which can be verified on the Type II example (1 unit of flow plus 1 unit of recycling provides 2 units of consumption). On the one hand, increasing recycling reduces the total amount of plastics in the flow as well as plastic pollution. On the other hand, minimizing the total flow makes it more difficult to satisfy the total consumption. We can design a binary search algorithm to find the optimal flow and optimal recycling value, as follows:1.Define upper and lower limits to the total flow $F_{u}=\mathrm{Total}\;\mathrm{Consumption}$, $F_{l}=0$.2.While $F_{u}>F_{l}$, define mid-point total flow $F_{m}=\mathrm{floor}\left(\frac{F_{u}+F_{l}}{2}\right)$.3.Find a maximum flow of *G* with minimum cost, subject to the total flow value does not exceed $F_{m}$; this can be done by adding an additional edge to the source or sink, setting its capacity to be $F_{m}$, and then using an out-of-the-box solver in polynomial time.4.If the flow satisfies Constraint III, assign $F_{u}\overset{}{\leftarrow }F_{m}$; otherwise, assign $F_{l}\overset{}{\leftarrow }F_{m}+1$.5.Return $F_{u}$ and maximum flow *f* for capacity $F_{u}$ when $F_{u}\leq F_{l}$.

#### 2.4.2. From Mismanaged Plastic Waste to Ocean and River Pollution

Our model computes the total cost of the network flow, which can reflect the total mismanaged plastic waste (MMPW) created or the MMPW that ends up polluting the ocean. The latter is often much smaller than the former because MMPW has a relatively low probability of ending up in oceans (Link for data download: https://ourworldindata.org/grapher/probability-mismanaged-plastic-ocean (accessed on 13 November 2021), [Our World in Data: Probability of mismanaged plastic waste being emitted to ocean, 2019]). We also note that the probability of MMPW polluting the ocean varies significantly across countries.

For each country, the total amount of plastic waste is $f(C_{i}\to D_{i})$, and we multiply the MMPW factor for each country to obtain the MMPW in each country. For total pollution in oceans by each country, we multiply per-country MMPW with the probability that MMPW will be emitted to oceans from [13]. For pollution in individual rivers, we use the river pollution model in [14,15] which accounts for both macroscopic and microscopic plastic; we take the sum of both types to estimate the pollution from each river, which is estimated to be linear to the total population surrounding the river times the MMPW per person around the river. These data are then used to produce the world map plots that show improvements in ocean pollution, river pollution, and MMPW in general.

## 3. Results

### 3.1. Exploratory Data Analysis

The correlations among the proportion of different kinds of waste are shown in the left graph of Figure 3. Asterisks suggest statistical significance after multi-hypothesis correction.

Specifically, plastic waste shows nominally significant positive correlation with paper or cardboard waste (*p*-value = 0.02). In the right graph of Figure 3, we divide countries into four income levels: low income (LIC), lower-middle income (LMC), upper-middle income (UMC), and high income (HIC) and show the association between income level and the proportions of different kinds of waste. In particular, the proportion of plastic waste in low-income countries is significant less than those of other countries (*t* test *p*-value = 0.002), relating the structure of waste generation to country income. We did not observe significant association between the plastic waste as the proportion of total waste and import/export status of the countries, as shown in Appendix A—Figure A1.

Next, we show that the income level of countries is also associated with *mismanaged* plastic waste per capita. In the left graph of Figure 4, we observed that the lower-middle income countries and upper-middle income countries generate more mismanaged plastic waster per capita (kilograms per person per day) than low-income countries and high-income countries due to their more rapid development in the industrial sector. In the right graph of Figure 4, we compare mismanaged plastic per capita with log-scaled GDP per capita. Similarly, we see an inverse-U curve pattern. Thus, helping the LMCs and UMCs to develop effective waste management systems including the ability to recycle waste and the import/export polices making is crucial to reduce plastic pollution.

Figure 5 shows the relationship between a country’s plastic goods’ trade (as a fraction of its total trade) and its GDP (or income level). Specifically, in the upper left graph, we see a significant positive correlation (*p*-value < 0.001) between a country’s plastic goods’ fraction in its export and its GDP per capita; in the upper right graph, however, we see a significant negative correlation (*p*-value < 0.001) between a country’s plastic goods’ fraction in its import and its GDP per capita. Those indicate that a high-income country tends to export the plastic products whereas a low-income country tends to import plastic products. This pattern holds stable across time, as shown in Appendix A—Figure A2. In terms of international trade on different forms of plastic, we define the export propensity of a specific form of plastic for each country as its export volume in US dollar over the sum of its import and export volume. As is shown in the lower left graph of Figure 5, which computes the correlation of the export propensity among the five forms of plastics, the export propensity of plastic waste is negatively correlated with the export propensity of all other forms of plastic. The asterisks represent statistical significance after Bonferroni correction. The negative correlation of export propensity of plastic waste reflects different import/export policies for different forms of plastic across the countries. For instance, the lower middle and left graphs of Figure 5 demonstrate that high-income countries export more plastic waste (*t* test *p*-value = 0.055) and import more final manufactured plastic goods (*t* test *p*-value <0.001) than countries with less income. To be specific, only 7 out of the 21 LICs are exporters of plastic waste while 43 out of 58 HICs are exporters of plastic waste. The net exports are the total exports per capita subtracted by the total imports per capita of the corresponding forms of plastic measured in US dollars at current price. This indicates that the HICs, which have more developed waste management system, tend to export their recyclable plastic waste to the countries with less income. We note that no matter what the form of plastic is, the import/export volumes per capita of HIC are consistently higher.

Figure 6 plots the relationship between a country’s oil production amount and its net plastic good export in 2010, using the countries that at least produced 100 TWh oil in that year. Given that oil is the most important input to produce plastic goods, not surprisingly, there is a strong positive correlation between the two (*p*-value = 0.003). This provides support to our model assumption that the capacity to produce raw form of plastic is correlated to the oil production amount (Stage I).

The left graph of Figure 7 presents the relationship between a country’s recycle rate of municipal solid waste and its GDP per capita level. Note that municipal solid waste is not the same as plastic waste, but the recycle rates of the two is perceived to be highly correlated [16]. Since we do not find data on plastic waste recycle rates, we use municipal solid waste recycle rates as proxies for that, as mentioned in Section 2.1. The figure shows that the recycle rates are strongly positively correlated with GDP per capita, indicating that richer countries tend to recycle more. This observation motivates our assumption in the model in terms of recycling capacity (Stage V, recycling via international trade). In addition, the right graph of Figure 7 shows that the mismanaged rate of plastic waste monotonously decreases with the income of the countries. The mismanaged plastic waste rate reflects the ability of waste management of the country. Better ability of waste management results in less mismanaged plastic waste. This is reasonable given that richer countries tend to have better waste management technology and higher regulatory standards. This observation is used in our model to specify the cost efficiency functions for each country (Stage IV, waste generation). To be specific, we used linear regression regressing the recycle rate on GDP per capita to impute the recycle rate for the countries without the data.

**Figure 5 ijerph-19-15963-f005:**
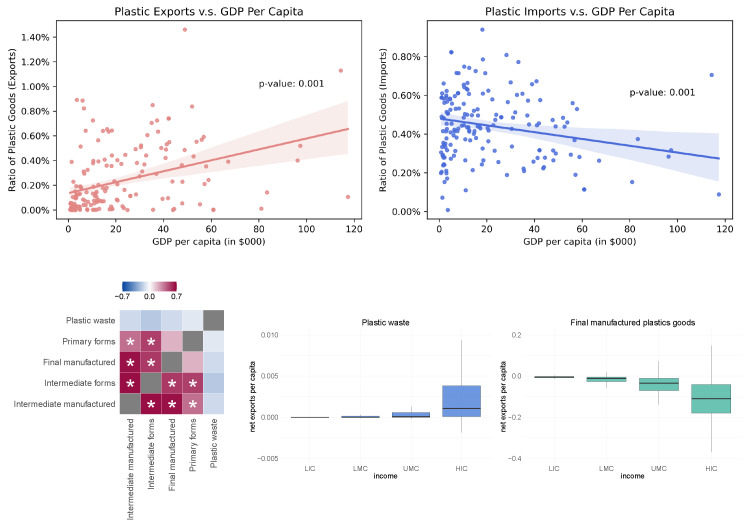
Plastic goods trade and countries’ GDP (or income level).

**Figure 6 ijerph-19-15963-f006:**
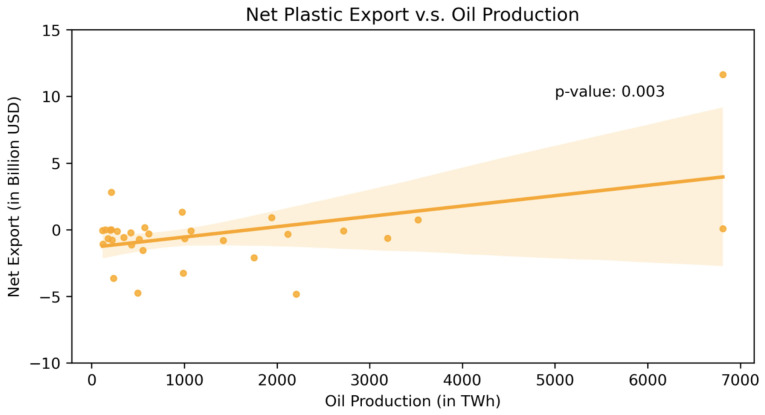
Plastic product net export and oil production.

**Figure 7 ijerph-19-15963-f007:**
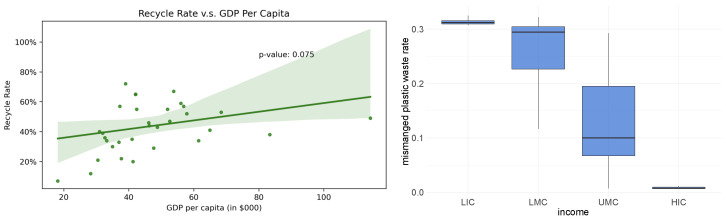
Recycle rate, waste mismanagement, and GDP per capita.

Put together, both the trade patterns and mismanaged waste generation of the countries are associated with country incomes. Developed countries, which show low mismanaged plastic waste rate, tend to export more plastic waste to the world. Meanwhile, we found significant positive association between oil production and raw plastic production, between GDP per capita and recycle rate, and between plastic waste export per capita and waste management ability which agree with all of our model assumptions.

### 3.2. Optimal Trade Pattern

With our network model being able to solve for an optimal trade flow, we are able to construct a trade "fairness" score to evaluate how close the empirical trade pattern in real world is to the optimal pattern produced by the model. As explained earlier, in this study, we mostly limit to the characterization of the trade of waste (while it could be easily generalized to other edges), and here, for each country, we compare the optimal to the empirical total imports from the world, as well as that with total exports to the world, respectively.

From Figure 8, it can be seen that the real-world empirical patterns positively correlate with the model-created optimal trade patterns (imports from the world and exports to the world for each country). The distances to the optimal trade pattern for most of the countries are actually small (Appendix A—Figure A3). Assuming the model-created pattern is optimal as a “ground truth”, the stronger the correlation is, the empirical trade flows are better. This allows us to define a metric to evaluate a given trade flow:(2)Trade Fairness Score = Corr(empirical trade flow,optimal trade flow)across countries
where the flow is represented as concatenated imports from the world and exports to the world vector. This metric helps people understand the fairness of trades in the context of a global economy rather than a “not in my backyard” thinking. For instance, a drastic trade policy change imposed by a single country may shift the trade patterns in other countries, and may hurt the global environment even if it is seemingly beneficial to its own economy.

## 4. Discussion

### 4.1. Policy Recommendations

In this section, we study the effects on ocean plastic pollution from varying two factors: (1) improving waste trade capacity, and (2) improving recycling rate. For each investigated factor, we control the other factor; this allow us to quantify the effect of modifying the factor that we care about. For each of the scenarios, we solve our network flow problem, and then use the solution to estimate the environmental impact of ocean plastic pollution from 316 rivers and from 176 countries. We estimate these quantities based on existing studies [13,14,15] (see Section 2.4.2 for more details).

#### 4.1.1. Increase Trade Limit for Plastic Waste

The first factor we investigate is the trade limit, where we fix the recycling capacities to be equal to estimates from historical data. We study three simple scenarios: **no trade** where trade capacity is equal to historical data and recycling capacities are zero (countries do not recycle others’ plastic waste); **trade** where trade capacity and recycling capacities are equal to historical data; **more trade** where trade capacity is further multiplied by 10.

Our solution to **trade** and **more trade** reduced global ocean plastic pollution by 2.5% (31,226 tons) and 14.0% (181,681 tons) over **no trade**, respectively. We demonstrate the improvements of **trade** and **more trade** policies over the **no trade** policy in Figure 9 and Figure 10. The baseline of **no trade** is shown in Appendix A—Figure A4. The additional improvements from more trade indicates the impacts of easing trade limits ([17]). First, we observe that none of the countries suffered from more contributions to global plastic pollution. Next, we observe that the amount of reduction in plastic pollution is not uniform over all countries. For example, Philippines is a country that has a high rate of producing plastic pollution to the ocean. Compared to **no trade**, **trade** decreases its ocean plastic pollution by 9.4% whereas **more trade** decreases its ocean plastic pollution by 91.7%; these results suggest that Philippines exports its plastic waste to other countries that recycle it and use it in production. In Figure 11, we illustrate the pattern of plastic waste exports for China and Philippines. Naturally, we observe that more export occurs with **more trade**. We show a similar figure for India and Mexico in Figure A5 in the Appendix A. All in all, these results suggest that we should increase the trade capacity for plastic waste trade if there is enough capacity to recycle these plastics for production.

#### 4.1.2. Increase Recycle Rate of Plastic Waste

We then test the effect of recycle rate on reducing pollution. We uniformly increased the recycle rate by 10% relative to a baseline model. It is not surprising that a better plastic recycling system should help with the pollution problem. Interestingly though, we observe a clear negative correlation between the ocean pollution reduction and GDP per capita across countries (Figure 12), and a similar pattern also exists with the previous trade limit strategy. This suggests that there is more space for improvement in developing countries, where the recycling management system is still largely under-developed [18]. Appendix A—Figure A6 and Figure A7 display the anticipated pollution reduction from increasing the recycle rate on a geographical map. This heterogeneity also suggests that some countries have larger policy levers. For instance, because China is the largest developing country, improvement in recycling rate in China alone can significantly reduce global ocean plastic pollution, compared to developed countries and small developing countries, consistent with previous studies [9,19].

### 4.2. Comparison with Literature and Future Directions

We have introduced a network flow model that encompasses the life-cycle of plastics in terms of production, consumption, disposal, recycling, and international trade. Given realistic constraints, we can solve for the optimal trade flows that minimize global plastic pollution. The optimal trade pattern given by our model is highly correlated to the empirical trade flows in plastic waste. We also show that increasing international trade of plastic waste and domestic plastic recycling rate are effective in terms of reducing plastic pollution. As plastic waste trade is only a minor component in international trade, such a policy is actionable with a globally coordinated effort. Our study shows that international cooperation between developed and developing countries is crucial to help raise the recycling efficiency, which would significantly alleviate the plastic pollution issue, consistent with existing proposals [20,21,22,23].

Compared to the existing literature that study the global (UN plastic treaty [8,9]) and local [24] plastic product flow, we add more mathematical structure to the problem and solve the model using the combination of model simulation and real-world data. The inner structure of the trade flow model allows us to capture the subtle impacts throughout the production line of the plastic products. In addition, the flexible choice for parameterization incorporates the possibility for wide application of the model. In terms of policy implications, our model not only provides *qualitative* policy suggestions, but also offers *quantitative* estimates for each policy impacts. Plus, the heterogeneity of policy impacts across the line of GDP per capita implied by our model is a unique and novel finding on top of the existing literature.

In this study, we only considered conventional fossil-based plastics, but not bio-plastics that can biodegrade at a much faster rate than conventional plastics [25]. This is because currently none of the commonly used plastics are biodegradable and do not completely disintegrate [26].

However, our model is not without its limitations. First, we assume that the cost function is shared across the globe and does not optimize for reducing plastic waste in a particular region; this causes some countries to have a larger reduction of plastic waste than others. Second, our study currently only focuses on the trade of plastic waste, but it can be easily generalized to limit other of trade. Third, the cost function used in our study currently does not explicitly consider economic costs such as shipping and tariff. Last but not least, our model does not consider the temporal effect of international trade on plastics recycling—even though we have the capacity to trade and recycle plastic does not mean it will be allocated to the right country in time, and we may need additional plastic production to take care of the overhead. Nevertheless, our model provides a first attempt at addressing plastic pollution via international trade, our suggested policies can be implemented in reality, and the proposed trade patterns may benefit both the environment and human consumption.

## 5. Conclusions

In this paper, we aim to offer insights in plastic waste management through a global trade flow perspective. Trade flows are key to understanding the global plastic market and the plastic product supply chain. To motivate our study, we first establish empirical facts of the distributional patterns for raw inputs, plastic products, and plastic waste. We then lay out a network flow model featuring the life-cycle of plastic in both domestic and international economic activities. Our model quantifies the global flows of production, consumption, and trade across the life-cycle of plastics. We find that the optimal flow determined by our model is highly correlated with reality, adding credibility to our model. For policy guidance, we use our model and simulation to point out two policy implications: increasing trade capacity of plastic waste reduces pollution; increasing plastic waste recycle rate reduces pollution and the effect is particularly strong for developing countries.

## Figures and Tables

**Figure 1 ijerph-19-15963-f001:**
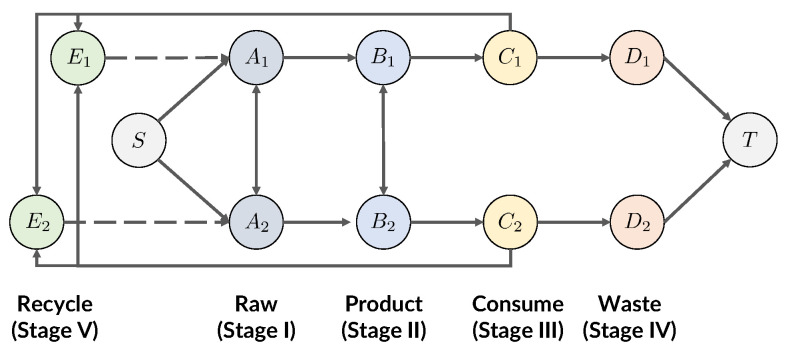
The plastic network flow model within the economy of two countries, including production (Stages I, II), consumption (Stage III), waste (Stage IV), and recycling (Stage V). Our model applies to more countries in the world and can be extended to more stages.

**Figure 3 ijerph-19-15963-f003:**
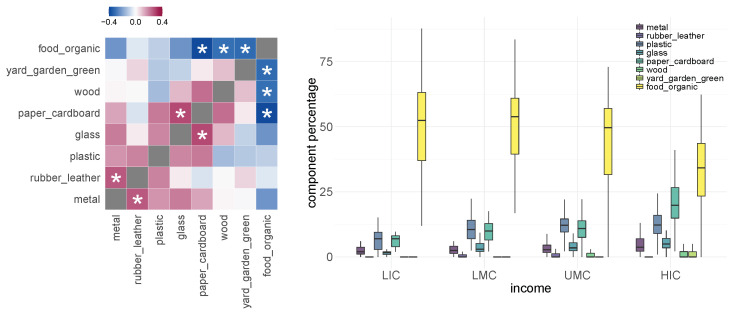
Correlation of proportions of different kinds of waste and their relation with country income.

**Figure 4 ijerph-19-15963-f004:**
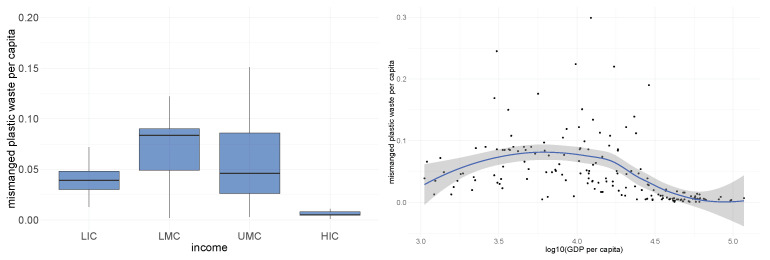
Plots of relationship between proportions of different kinds of waste and country income.

**Figure 8 ijerph-19-15963-f008:**
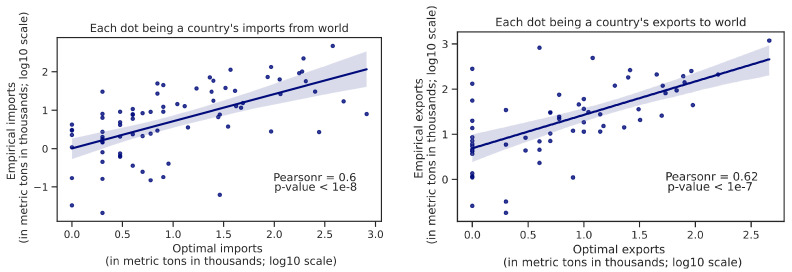
Comparison between the plastic waste trade pattern in reality and in optimal allocation (**Left**: imports from the world; **Right**: exports to the world).

**Figure 9 ijerph-19-15963-f009:**
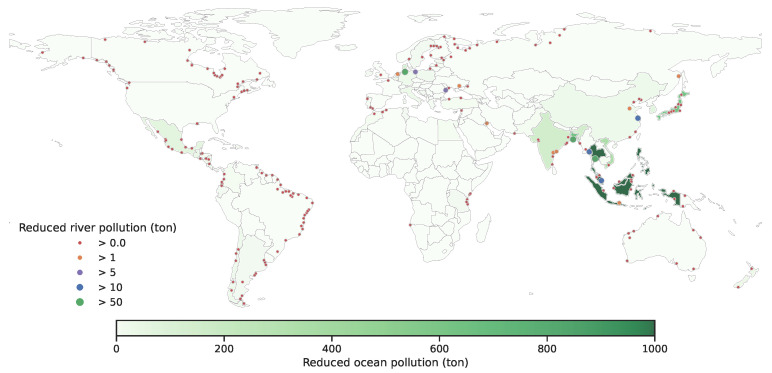
Improvements (in tons) in ocean and river pollution of **trade** over **no trade**.

**Figure 10 ijerph-19-15963-f010:**
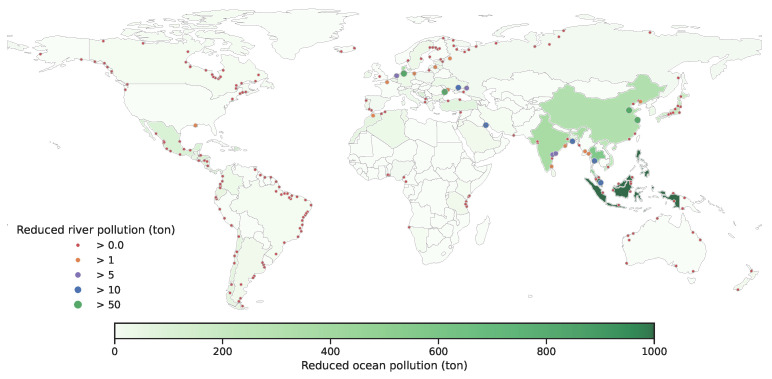
Improvements (in tons) in ocean and river pollution of **more trade** over **no trade**.

**Figure 11 ijerph-19-15963-f011:**
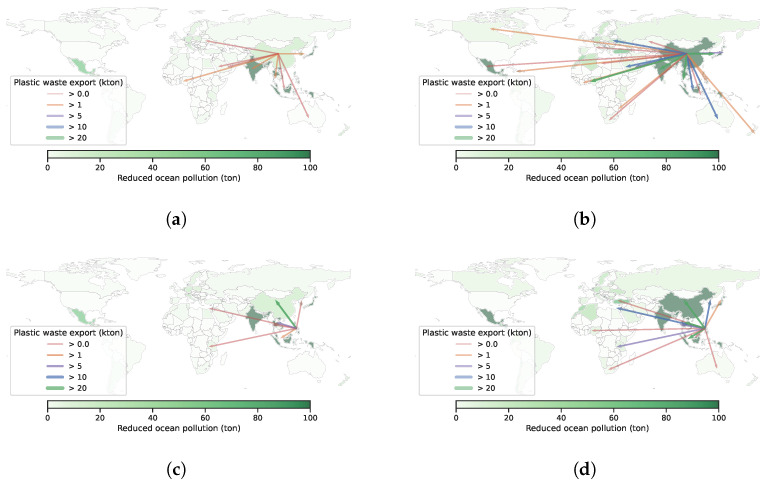
Export of China and Philippines’ plastic waste under different policies on global plastic waste trade. (**a**) **trade**, China. (**b**) **more trade**, China. (**c**) **trade**, Philippines. (**d**) **more trade**, Philippines.

**Figure 12 ijerph-19-15963-f012:**
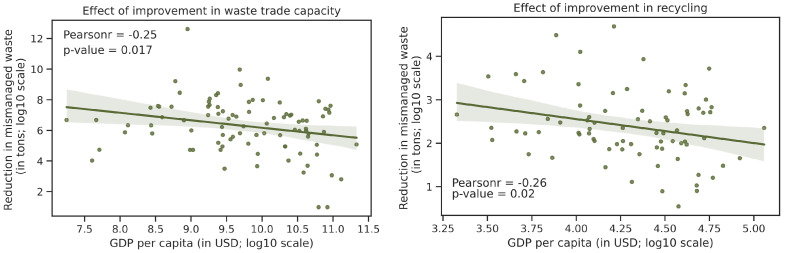
Effectiveness of policies in pollution reduction and countries’ GDP per capita (**Left**: increase of trade limit; **Right**: increase of recycle rate).

## Data Availability

Throughout the paper, the data used in the analysis have links in the footnote providing where to download them. We have no private data usage and all the data used are publicly available online.

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
