# Peer review of "Trade Flow Optimization Model for Plastic Pollution Reduction"

_ijerph, 2022, doi:10.3390/ijerph192315963_

Round 1
Reviewer 1 Report
Thank you for allowing me to read your paper ‘Trade Flow Optimization Model for Plastic Pollution Reduction.`
The paper is well-written; however, I have some recommendations, which I will detail in the following:
- Introduction: Please insert explicitly the research aim before explaining how you intend to address the problem
- Section 2. Material and Method: in the expected Method sub-section, which seems to miss, you must explain how you obtained and analyzed your results
- I recommend reorganizing Section 3. Results and Section 4. Discussion. Readers should expect to find in the 3. Results section, the key results of your research, and at the end of the 4. Discussion section policy implications based on your model. Maybe some conclusions should be extended, especially in comparison with prior studies.
I hope these comments are helpful when revising the manuscript, and I wish you all the best for your future research in this area.
Author Response
Dear reviewer,
Please see below for our point by point response to your insightful comments.
- Introduction: Please insert explicitly the research aim before explaining how you intend to address the problem
This is a really valuable suggestion to improve the clarity of our paper. We now have rewritten the first paragraph of the introduction, to make our research aim and related background information explicit. There we highlight the challenges and complexity of the problem, what the current gap is, and what we want to achieve regarding what we think is missing in the field.
- Section 2. Material and Method: in the expected Method sub-section, which seems to miss, you must explain how you obtained and analyzed your results
We agree on this point and are happy to make the change. Previously the technical details were placed in the results section, which was not the correct place for the ease of reading. We have now moved our quantitative modeling workflow to the method subsection, which should make the paper more organized and easier to read.
- I recommend reorganizing Section 3. Results and Section 4. Discussion. Readers should expect to find in the 3. Results section, the key results of your research, and at the end of the 4. Discussion section policy implications based on your model. Maybe some conclusions should be extended, especially in comparison with prior studies.
This is a great suggestion. We reorganize the results section and put part of the previous Section 3 (model) into Methods and put the policy implications into Section 4. We also compare our policy implications with the findings in the literature and particularly with the UN treaty paper suggested by the editor and a recent MDPI publication.
Reviewer 2 Report
Summary: This paper analyses the management of plastic waste worldwide. It considers the trade flows of the global plastics market between different countries and poses an optimisation problem to reduce plastic pollution in the oceans. The main conclusion argues for increasing plastics trade capacity and improving recycling rates in developing countries.
General comments:
This paper is very interesting and presents a very topical problem from a novel perspective. How to reduce plastic waste pollution in the oceans by optimising the flow of plastic between countries.
For this purpose, the authors have proposed a directed graph model that includes more than 170 countries where three states in the plastic life process are considered.
The paper is correct and generally well-written. The conclusions seem to be correct based on the data used.
Some general issues to be considered before the publication of the paper would be:
1) Indicate that all the data used to refer to the year 2010. In this sense, some reference should be made to how these data have changed up to the present day.
2) Given that we do not have plastic recycling rates for all countries, data from 32 countries have been considered and a correlation has been established between this rate and the GDP of the country, using this model to estimate rates for the remaining countries. A theoretical and practical justification for this estimation should be made.
3) In the analysis of the network, two fundamental variables are considered: the capacity and the cost of each arch. The paper does not explain how these functions are established.
4) Define in more detail the rate of creating mismanaged plastic (MMPW).
5) When we use total flow in the edge we are considering export or import trade. Define "total flow" in the paper and its relation to export/import trade.
6) How do we measure the impact of increasing the plastic trade limit between countries on the reduction of plastic pollution in the oceans?
Another important issue would be to incorporate appendices A.1 and A.2 into the text, as this would help to resolve some of the doubts raised in the previous points.
Finally, we consider that the text would be improved if you incorporate the figures in Appendix B, which are cited in the text, and delete the rest of the appendix. Some figures are duplicated in the text and Appendix B (i.e. Figures B5 and B6).
Specific comments:
Page 2, line 67: type mistake (co untries)
Page 4, line 143: Number of Figure omitted (??)
Page 4, line 149: This might be “Bonferroni correction”.
Page 9, line 305: Too long sentence.
Page 10, line 330: Add Appendix 2 to (Figure B2)
Page 11, line 373: Add Appendix 2 to (Figure B7)
Page 11, line 374: Add Appendix 2 to (Figure B8)
Author Response
Dear reviewer,
Please see our point by point reply below.
1) Indicate that all the data used to refer to the year 2010. In this sense, some reference should be made to how these data have changed up to the present day.
Thank you for the comment. It is true that the data being used changes over years. For example, the total global plastics production keeps increasing over the years, and the mismanaged plastics waste is also projected to double by 2025 (Jambeck et al. 2015). Still, each country’s share in plastics production, consumption, and recycling remain at similar levels. While our model is constrained using 2010 data, the framework is applicable to any arbitrary year. We have added this comment with reference to the subsection 2.1 Data.
2) Given that we do not have plastic recycling rates for all countries, data from 32 countries have been considered and a correlation has been established between this rate and the GDP of the country, using this model to estimate rates for the remaining countries. A theoretical and practical justification for this estimation should be made.
Thank you for your comment. In section 3.1 and Figure 7, we have shown the positive correlation between GDP per capita and recycle rate of municipal solid waste which is used as a proxy for the recycle rate of plastic waste. In another analysis in the same paragraph of the main text, we also demonstrated the negative correlation between countries’ income and the mismanaged rate of plastic waste. We have added more clarifications in the revised manuscript.
3) In the analysis of the network, two fundamental variables are considered: the capacity and the cost of each arch. The paper does not explain how these functions are established.
Thank you for your comment. In section 2.1, we have explained that the capacity between country A and country B in the network is the average trade volume of the corresponding form of plastic between country A and country B. The cost is defined as the mismanaged plastic waste rate. We have made it more clear in our revised manuscript.
4) Define in more detail the rate of creating mismanaged plastic (MMPW).
Thank you for the comment. In section 2.3, we have defined the rate of creating mismanaged plastic, which is defined by mismanaged plastic waste generation of country i / plastic waste generation of country i.
5) When we use total flow in the edge we are considering export or import trade. Define "total flow" in the paper and its relation to export/import trade.
Thanks for the question, we define it in Section 2.2.2. The term “total flow” refers to the amount of plastic products that are not recycled. It does not refer to export or import alone.
6) How do we measure the impact of increasing the plastic trade limit between countries on the reduction of plastic pollution in the oceans?
Thanks for the question – to clarify, we are measuring the impact of increasing the limit but first examining the difference between trade vs no trade and then see how that difference is changed when we increase the limit and introduce more trade. Basically, we treat the marginal impact as the impact coming from increasing the trade limit.
Another important issue would be to incorporate appendices A.1 and A.2 into the text, as this would help to resolve some of the doubts raised in the previous points.
Thanks for the suggestion. I have moved this to 2.4 in the model part, which fits nicely.
Finally, we consider that the text would be improved if you incorporate the figures in Appendix B, which are cited in the text, and delete the rest of the appendix. Some figures are duplicated in the text and Appendix B (i.e. Figures B5 and B6).
Thanks for the comments. The original Figure B5 and B6 are actually slightly different from Figure 9 and Figure 10 because the two appendix figures are reflecting the numbers in % instead of in tons. But given that these two figures do not add much additional value and are introducing confusion, we take the two appendix figures out.
Specific comments:
Page 2, line 67: type mistake (countries) Fixed
Page 4, line 143: Number of Figure omitted (??) Fixed
Page 4, line 149: This might be “Bonferroni correction”. Fixed
Page 9, line 305: Too long sentence. Fixed
Page 10, line 330: Add Appendix 2 to (Figure B2) Fixed
Page 11, line 373: Add Appendix 2 to (Figure B7) Fixed
Page 11, line 374: Add Appendix 2 to (Figure B8) Fixed